# Pulsatilla Saponin D Suppresses Proliferation and Induces Apoptosis in Human Prostatic Cells

**DOI:** 10.3390/cells14211706

**Published:** 2025-10-30

**Authors:** Yuzhong Chen, Ping Zhou, Yangtao Jin, Dongyan Huang, Xin Su, Congcong Shao, Juan Jiang, Rongfu Yang, Jianhui Wu

**Affiliations:** 1NHC Key Lab of Reproduction Regulation, Shanghai Engineering Research Center of Reproductive Health Drug and Devices, Shanghai Institute for Biomedical and Pharmaceutical Technologies, Pharmacy School, Fudan University, Shanghai 200237, China; 24211150006@m.fudan.edu.cn (Y.C.); zhouping19a@163.com (P.Z.); jyt_sphu@163.com (Y.J.); hdy043@163.com (D.H.); suxiaoxin1982@163.com (X.S.); shaocongcongscc@163.com (C.S.); jiangjuan06@126.com (J.J.); yangrongfu82@163.com (R.Y.); 2Department of Pharmacology & Toxicology, Shanghai Institute for Biomedical and Pharmaceutical Technologies, Shanghai 200032, China

**Keywords:** triterpenoid saponin, Pulsatilla saponin D, prostatic hyperplasia

## Abstract

The growing global aging population is contributing to an increasing burden of benign prostatic hyperplasia (BPH), highlighting the need for novel, highly effective and low-toxicity therapies. In light of its well-documented anti-inflammatory and anti-tumor properties, we investigated the potential of the natural product Pulsatilla saponin D (PSD) in treating BPH. For the first time, we demonstrate that PSD significantly inhibits the proliferation of and induces apoptosis in the immortalized human normal prostatic stromal cell line, human prostate fibroblasts, and the human benign prostatic hyperplasia epithelial cell line. Mechanistic studies involving transcriptome analysis and RT-qPCR validation revealed that PSD likely exerts its effects by downregulating the expression of the androgen receptor and by modulating multiple signaling pathways synergistically, including the Phosphatidylinositol 3-kinase/Protein Kinase B, Tumor Necrosis Factor, Hypoxia-Inducible Factor-1 and Interleukin-17 pathways.

## 1. Introduction

Benign prostatic hyperplasia (BPH) refers to the non-cancerous growth or hyperplasia of prostate tissue. It is a common cause of lower urinary tract symptoms in aging men. The condition is characterized by the proliferation of stromal and epithelial cells within the transitional zone of the prostate. The prevalence of the disease has been shown to increase with age, with histopathological incidence rates reaching 50–60% in men aged 60 and rising to 80–90% in those over 70 years old [1]. Consequently, as human life expectancy increases and greater emphasis is placed on quality of life, future patients with BPH are likely to have higher expectations of therapeutic outcomes [2].

Currently, the therapeutic options for patients with symptomatic BPH range from behavioral modifications and pharmacological interventions to minimally invasive surgical procedures [3]. Prostate growth development depends on androgen stimulation. Specifically, testosterone is converted by the enzyme 5α-reductase into dihydrotestosterone (DHT) within the stromal and basal cells of the prostate. DHT is a more potent androgen which subsequently drives excessive prostate cell proliferation [4]. α-Adrenergic receptor blockers aim to minimize muscle tone in the smooth muscle of the prostatic stroma and the bladder neck tissue. Agents such as tamsulosin and doxazosin, which are α1-blockers, antagonize this receptor, effectively reducing the tension of the smooth muscle surrounding the prostatic urethra and alleviating obstructive symptoms caused by BPH [5,6,7]. 5α-Reductase inhibitors, including finasteride and dutasteride, block the conversion of testosterone to DHT, thereby reducing prostate volume [8]. However, the long-term use of these medications is often associated with a range of adverse effects such as orthostatic hypotension and sexual dysfunction. Furthermore, tolerance to some drugs may develop with prolonged administration, resulting in a gradual decline in their efficacy over time [9]. Consequently, developing novel therapeutic agents for BPH that are highly effective yet less toxic is a critical area of current research.

Natural products, derived from animal and plant extracts, insects, marine organisms, or microbial compounds, constitute an important component of drug discovery [10]. These compounds exhibit diverse pharmacological activities, including anti-inflammatory, antioxidant, and immunomodulatory effects, and are capable of simultaneously targeting multiple potential therapeutic pathways [11]. Triterpenoid saponins are a significant class of plant secondary metabolites, classified as glycosides composed of a triterpenoid aglycone and a sugar chain linked via glycosidic or ester bonds. These saponins are found in plants such as Bupleurum, Panax notoginseng, and Platycodon grandiflorus, and demonstrate biological activities including inhibition of cell proliferation, induction of apoptosis, anti-invasion, and immunoregulation. Ginsenoside Rh2, a specific triterpenoid saponin, inhibits the proliferation and invasion of prostate cancer cells in both in vitro and in vivo models [12]. This effect is mediated through the activation of the transforming growth factor beta receptor and subsequent regulation of key protein expression, such as the downregulation of cyclin D1, cyclin B1, matrix metalloproteinase-2 (MMP-2), and MMP-9 [13]. Furthermore, certain saponins can enhance the antitumor efficacy of chemotherapeutic agents. For example, Panax notoginseng saponins act as chemosensitizers by enhancing gap junction function through connexin 32 and 26, thereby promoting the cytotoxic effects of cisplatin [14]. Additionally, dammarane-type saponins, such as protopanaxadiol, primarily enhance the activity of caspase-3 and caspase-9, resulting in pro-apoptotic interactions with chemotherapeutic drugs like irinotecan in human colorectal cancer cell lines (e.g., HCT-116 and SW-480) [15].

Pulsatilla saponin D (PSD), an oleanane-type pentacyclic triterpenoid extracted from the roots of Pulsatilla chinensis, exhibits a broad spectrum of pharmacological activities and demonstrates considerable potential in antitumor applications (Figure 1) [16,17]. Recent studies have further elucidated its diverse mechanisms of action. Jin et al. reported that PSD inhibits osteosarcoma growth by modulating the JNK/ATF3 signaling pathway, a mechanism distinct from those previously described [18]. Furthermore, Jang et al. found that PSD specifically suppresses phosphorylation of c-Met and its downstream Akt signaling, effectively restraining the growth of gefitinib-resistant non-small cell lung cancer harboring *c-Met* amplification [19]. Additionally, Lin et al. demonstrated that PSD enhances the sensitivity of BRCA1/2 wild-type ovarian cancer to PARP inhibitors by inhibiting BRCA1 protein function [20]. Therefore, we consider it meaningful to investigate whether PSD plays a role in suppressing the abnormal proliferation of prostate cells.

In this study, we investigated the anti-proliferative effects of PSD on prostate cells through a series of in vitro experiments, including assessments of cell proliferation and apoptosis. Doxazosin (40 μM) was used as a positive control. Mechanistically, we hypothesized that PSD exerts its anti-proliferative activity by targeting inflammatory and apoptotic pathways. Transcriptomic analysis was employed to identify potential signaling pathways, followed by validation of key genes at the mRNA level. This research demonstrates that PSD is a promising natural agent candidate for the treatment of BPH.

## 2. Materials and Methods

### 2.1. Chemicals

PSD (CAS No. 68027-15-6, purity: 99.89%) was purchased from MedChemExpress (Shanghai, China) and dissolved in dimethyl sulfoxide (DMSO). DMSO was obtained from Thermo Fisher Scientific (Shanghai, China). Doxazosin (CAS No. 74191-85-8, purity: ≥98%) was sourced from Macklin Biochemical Co., Ltd. (Shanghai, China). All reagents were diluted in the corresponding culture media as described in Section 2.2. The Cell Counting Kit-8 (CCK-8) and the Annexin V-FITC/PI Apoptosis Detection Kit were supplied by Beyotime Biotechnology (Shanghai, China) and Yeason (Shanghai, China), respectively.

### 2.2. Cell Culture

The immortalized human normal prostatic stromal cell line (WPMY-1) was purchased from Shanghai Zhongqiao Xinzhuo Biotechnology Co., Ltd. (Shanghai, China). Human prostate fibroblasts (HPRF) were obtained from Sciencell Research Laboratories (Shanghai, China). The human benign prostatic hyperplasia cell line (BPH-1) was acquired from Shanghai Yansheng Biotechnology Co., Ltd. (Shanghai, China). WPMY-1 cells were cultured in high-glucose DMEM medium supplemented with 5% fetal bovine serum (FBS). HPRF cells were maintained in FM medium containing 2% FBS and 1% fibroblast growth supplement. BPH-1 cells were grown in F12K medium with 10% FBS. All cell lines were incubated at 37 °C in a humidified atmosphere with 5% CO_2_, and all culture media were supplemented with 1% penicillin-streptomycin solution.

### 2.3. Cell Morphology

Cells were seeded in 96-well plates at a density of 4000 cells per well for WPMY-1, HPRF, and BPH-1 cell lines, respectively. The cells were then exposed to PSD (2–8 μM) or doxazosin (40 μM). After 48 h of treatment, cell morphology was observed and photographed under a microscope.

### 2.4. Cell Viability Assay

Cells were seeded in 96-well plates at densities of 4000 cells per well (for 48 h treatment) or 3000 cells per well (for 72 h treatment). The cells were then treated with various concentrations of PSD (500 nM to 8 μM) or doxazosin (40 μM) in the culture medium for 48 or 72 h. Following treatment, CCK-8 solution was added to each well, and the plates were incubated at 37 °C for 1–4 h protected from light. The absorbance was measured at a wavelength of 450 nm using a microplate reader (Biotek, Santa Clara, CA, USA).

### 2.5. Annexin V-FITC/PI Double Staining Apoptosis Detection

Cells were seeded in 6-well plates at a density of 2.5 × 10^5^ cells per well and cultured overnight. Subsequently, the cells were treated with PSD (500 nM to 4 μM) or doxazosin for 48 h. After treatment, the cells were digested, centrifuged, and resuspended in 1× Binding Buffer (Yeason, Shanghai, China). Annexin V-FITC and PI staining solutions were added to the cell suspensions, which were then incubated at room temperature for 10–20 min protected from light. Finally, the samples were analyzed using a flow cytometer (BD Biosciences, San Jose, CA, USA).

### 2.6. Transcriptome Sequencing

Total RNA was extracted from WPMY-1 and BPH-1 cells treated with 2 μM PSD using TRIzol reagent, followed by removal of genomic DNA with DNase I (Takara, Kyoto, Japan). The RNA samples were subjected to RNA sequencing at Lingbio Biotechnology Co., Ltd. (Shanghai, China). Differentially expressed genes (DEGs) were identified with the screening criteria of |log2(fold change, FC)| ≥ 2 and false discovery rate (FDR) ≤ 0.05. Gene Ontology (GO) and Kyoto Encyclopedia of Genes and Genomes (KEGG) enrichment analyses were performed based on the DEGs, and the results were visualized using scatter plots and volcano plots.

### 2.7. Real-Time Quantitative Polymerase Chain Reaction (RT-qPCR)

Total RNA was isolated from cells using the RNAeasy™ Kit (R0026, Beyotime Biotechnology, Shanghai, China). The purity and concentration of the RNA were determined by a spectrophotometer (ACT Gene, Shanghai, China). Primer sequences were designed using Primer Premier 6 software (details provided in Table 1). After reverse transcription, the mRNA expression levels were detected on a Roche LightCycler^®^ 480 quantitative PCR system (Roche, Rotkreuz, Switzerland), and the relative expression of target genes was calculated using the 2−ΔΔCt method.

### 2.8. Statistical Analysis

All quantitative data are expressed as the mean ± standard deviation (SD). Statistical analyses and graph generation were performed using the SPSS statistical software package (version 26.0) and GraphPad Prism (version 8.0; GraphPad Software, San Diego, CA, USA). A *p*-value of less than 0.05 was considered statistically significant.

## 3. Results

### 3.1. Effect of PSD on the Morphology of WPMY-1, HPRF, and BPH-1 Cells

After 48 h of treatment with PSD (2, 4, and 8 μM), concentration-dependent morphological alterations were observed in WPMY-1, HPRF, and BPH-1 cells (Figure 2). These changes included reduced cell number, shrinkage, rounding, and decreased adhesion capacity. Furthermore, the sensitivity of the three cell lines to PSD varied, with the order of sensitivity being HPRF > WPMY-1 > BPH-1.

### 3.2. PSD Inhibits the Growth of WPMY-1, HPRF, and BPH-1 Cells

The inhibitory effects of PSD on the growth of WPMY-1, HPRF, and BPH-1 cells were evaluated using the CCK-8 assay. As shown in Figure 3, treatment with 1–8 μM PSD for 48 h significantly inhibited the viability of WPMY-1 cells, while PSD at concentrations ranging from 500 nM to 8 μM markedly suppressed the growth of HPRF cells. After 72 h of treatment, 8 μM PSD significantly reduced the viability of BPH-1 cells. In summary, PSD suppressed the growth of all three cell lines in a dose- and time-dependent manner.

Following 48 h of treatment, the half-maximal inhibitory concentrations (IC50) for WPMY-1, HPRF, and BPH-1 cells were 2.649 μM, 1.201 μM, and 4.816 μM, respectively. After 72 h of treatment, the IC50 values for these cell lines were 2.511 μM, 1.192 μM, and 4.315 μM, respectively (Table 2).

### 3.3. PSD Induces Apoptosis in WPMY-1, HPRF, and BPH-1 Cells

To further investigate the effect of PSD on prostate cells, apoptosis was assessed using Annexin V-FITC/PI staining followed by flow cytometry. The results showed that treatment with PSD at concentrations ranging from 500 nM to 4 μM significantly increased the apoptosis rate in WPMY-1 cells (Figure 4A,B). Similarly, exposure to 1–4 μM PSD induced a significant increase in apoptosis in HPRF cells (Figure 4C,D). Furthermore, 4 μM PSD demonstrated a stronger pro-apoptotic effect than 40 μM doxazosin in both WPMY-1 and HPRF cells. In BPH-1 cells, significant apoptosis induction was observed following treatment with 2 μM PSD (Figure 4E,F).

### 3.4. Transcriptomic Analysis Reveals PSD Modulation of Signaling Pathways in WPMY-1 and BPH-1 Cells

To elucidate the potential mechanism by which PSD inhibits prostate cell proliferation, we employed RNA-seq to assess gene expression profiles. Sample correlation analysis confirmed distinct clustering of gene expression profiles under different treatment conditions (Figure 5A,D). We screened DEGs, and scatter plots and volcano plots visualized the DEGs between WPMY-1 cells and the control group (Figure 5B,C); BPH-1 cell groups were visualized in a similar manner (Figure 5E,F). In the WPMY-1 cell line, 1202 genes were differentially expressed in the 2 μM PSD treatment group compared with the control group, comprising 754 upregulated and 448 downregulated genes. In the BPH-1 cell line, 1362 genes were differentially expressed, with 526 upregulated and 836 downregulated.

The Venn diagram shows 196 co-upregulated genes and 116 co-downregulated genes in the comparison between the two groups (Figure 6A–C). GO and KEGG enrichment analyses were performed on the intersections of differentially expressed genes (co-upregulated or co-downregulated) from both groups, respectively (Figure 6D,E). The co-upregulated genes were significantly enriched in signaling pathways such as TNF (Tumor Necrosis Factor), IL-17 (Interleukin-17), and HIF-1 (Hypoxia-Inducible Factor-1), while the co-downregulated genes were significantly enriched in pathways including PI3K/AKT (Phosphatidylinositol 3-kinase/Protein Kinase B) (Figure 7). The differentially expressed genes co-regulated by the above pathways included *TNC*, *IL1B*, *IL6*, *MMP3*, *CEBPB*, *FOSL1*, and *JUN* (Figure 8).

### 3.5. Regulation of AR Gene Expression by PSD

To determine whether PSD treatment affects the relative mRNA expression level of the androgen receptor (AR), we performed RT-qPCR assays in WPMY-1 and HPRF cells. As shown in Figure 9A,B, the expression of *AR* was significantly downregulated in WPMY-1 cells, whereas no significant effect was observed in HPRF cells.

## 4. Discussion

This study investigated the anti-proliferative effects of PSD on human prostate cells, shedding light on the underlying mechanisms. The results showed that PSD inhibited the proliferation of both stromal and epithelial prostate cells and induced apoptosis. These effects were mediated, at least in part, via the PI3K/AKT, TNF, HIF-1, and IL-17 signaling pathways.

Phosphatidylinositol 3-kinase acts as a central regulator of multiple intracellular signaling pathways and plays a critical role in fundamental cellular processes, including growth, apoptosis, and metabolism. The downstream target kinase AKT, along with phosphoinositide-dependent kinase 1 (PDK1), is recruited to the plasma membrane through its pleckstrin homology domain. Complete activation of AKT depends on phosphorylation at two key residues: threonine 308 by PDK1 and serine 473 by the mammalian target of rapamycin complex 2. Once activated, AKT regulates a broad range of biological functions—such as embryonic development, cell proliferation, nutrient uptake, anabolic metabolism, and cell survival and growth—by phosphorylating a series of substrate proteins [21].

The PI3K/AKT/mammalian target of rapamycin (mTOR) signaling pathway represents a therapeutic target for hormone-related diseases, including breast and ovarian cancer [22,23]. Concurrently, PI3K/AKT-phosphorylated Glycogen Synthase Kinase-3 Beta promotes tumor proliferation associated with multidrug resistance and is often considered a critical node regulating cancer multidrug resistance [24]. Furthermore, the PI3K/AKT signal can directly or indirectly induce the epithelial–mesenchymal transition by modulating the expression of Twist, Snail, and E-cadherin, or through crosstalk with other signaling pathways such as TGF-β, NF-κB, Rat Sarcoma, and Wnt/β-catenin, ultimately promoting tumor cell migration and invasion [25]. In prostate cancer, genes within the PI3K signaling pathway frequently undergo mutations, altered expression, or copy number changes. Loss of Phosphatase and Tensin Homolog and aberrant activation of the PI3K/AKT pathway further induce cholesteryl ester accumulation, strongly promoting prostate cancer proliferation and invasion [26,27]. Meanwhile, PI3K signal transduction interacts with Wnt, Mitogen-Activated Protein Kinase (MAPK), and AR signaling cascades, synergistically driving prostate cancer growth and therapy resistance, indicating that pathway inhibitors hold broad application potential in prostate cancer [28]. In this study, multiple genes related to PI3K/AKT signaling showed differential expression, including Tenascin C (*TNC*), Collagen Type I Alpha 1 Chain (*COL1A1*), Integrin Subunit Alpha V (*ITGAV*), and Platelet-Derived Growth Factor C (*PDGFC*), among others. *TNC*, an extracellular matrix component often considered a potential biomarker for cancer-associated fibroblasts, has been shown upon knockdown to effectively inhibit glucose uptake, lactate production, and the expression of key glycolytic enzymes in prostate cancer cells; clinical data analysis further confirms that *TNC* is significantly associated with poor prognosis in prostate cancer patients [29]. Additionally, *TNC* promotes epithelial–mesenchymal transition in nasopharyngeal carcinoma and activates the PI3K/AKT/mTOR signaling pathway, collectively supporting its role as an oncogene [30]. Similarly, *ITGAV*, *PDGFC*, *PDGFD*, and *COL1A1*—the most abundant collagen in human tissues—are often highly expressed in tumor cells, and inhibiting their expression can effectively suppress cell proliferation, migration, and invasion [31,32].

HIF-1 serves as a central transcriptional regulator mediating cellular responses to hypoxia, playing a pivotal role in maintaining cellular and organ homeostasis by orchestrating the expression of a series of downstream genes [33]. TNF and IL-17 signaling pathways collectively participate in modulating cytoskeletal dynamics and inflammatory mediator release, thereby contributing to cellular morphological regulation and inflammatory responses [34,35]. Key participants in these processes include inflammation-associated genes (e.g., *IL-6*, *IL-1β*), *MMP3*, CCAAT enhancer-binding protein beta, and natriuretic peptide A, which are involved in inflammation, immune responses, differentiation, and proliferation processes.

Son et al. reported that PSD inhibits the PI3K/AKT/mTOR signaling pathway, thereby inducing apoptosis, suppressing proliferation and angiogenesis, and demonstrating stronger anti-colon cancer activity than 5-fluorouracil in both in vitro and in vivo models [36]. PSD acts as an autophagy flux inhibitor in various cancer cells and induces autophagosome initiation; it can serve as a chemosensitizer, where its inhibition of mTOR and phosphorylation of its downstream target p70S6K induces autophagosome formation [37]. Substantial additional evidence indicates that the anti-proliferative potential of triterpenoid saponins is associated with the PI3K/AKT pathway. For instance, Momordin Ic can modulate oxidative stress and induce apoptosis in hepatocellular carcinoma cells via the MAPK and PI3K/AKT-mediated mitochondrial pathway, a process dependent on the promotion of heme oxygenase-1 and inducible nitric oxide synthase expression [38]. Afrocyclamin A inhibits the phosphorylation of PI3K, Akt, and mTOR in prostate cancer DU-145 cells, induces autophagic vacuole formation, and suppresses cancer cell migration and invasion in a dose-dependent manner [39].

On the other hand, certain triterpenoid saponins can effectively inhibit the expression of the *AR* gene or protein. For example, Hederacoside A, an oleanane-type saponin extracted from Anemone raddeana roots, induces proteasome-mediated degradation and inhibits the transcriptional activity of both full-length and splice variant AR, thereby suppressing prostate cancer cell growth; moreover, its combination with the chemotherapeutic drug docetaxel enhances therapeutic efficacy [40]. Ginsenosides inhibit the expression of AR, prostate-specific antigen (PSA), proliferating cell nuclear antigen, and 5α-reductase [41,42]. Holothurin A suppresses the growth of androgen-sensitive prostate cancer cells by reducing AR and PSA activity; molecular docking simulations reveal that it forms hydrophobic interactions with valine 676, isoleucine 680, and alanine 721 within the ligand-binding domain of AR, forming a stable complex [43]. The regulation of AR expression and transcriptional activity by PSD requires further clarification in future studies. Furthermore, studies using a BPH rat model will be conducted to further investigate the in vivo efficacy of PSD in alleviating BPH. These investigations will help clarify the pharmacological actions of PSD and its specific impacts on the relevant pathways and proteins.

## 5. Conclusions

Studies suggest that PSD may inhibit the proliferation of prostate cells (e.g., WPMY-1, HPRF, and BPH-1) by modulating the gene expression of the *AR*. This process potentially involves alterations in the expression of transcription factors and related proteins within signaling pathways such as PI3K/AKT, TNF, HIF-1, and IL-17, which collectively mediate the anti-proliferative effects of PSD on prostate cells. With ongoing research, we plan to validate key findings at the protein level, focusing on the PI3K/AKT pathway and the identified differentially expressed genes. PSD is a promising candidate for natural drugs used in the treatment of BPH.

## Figures and Tables

**Figure 1 cells-14-01706-f001:**
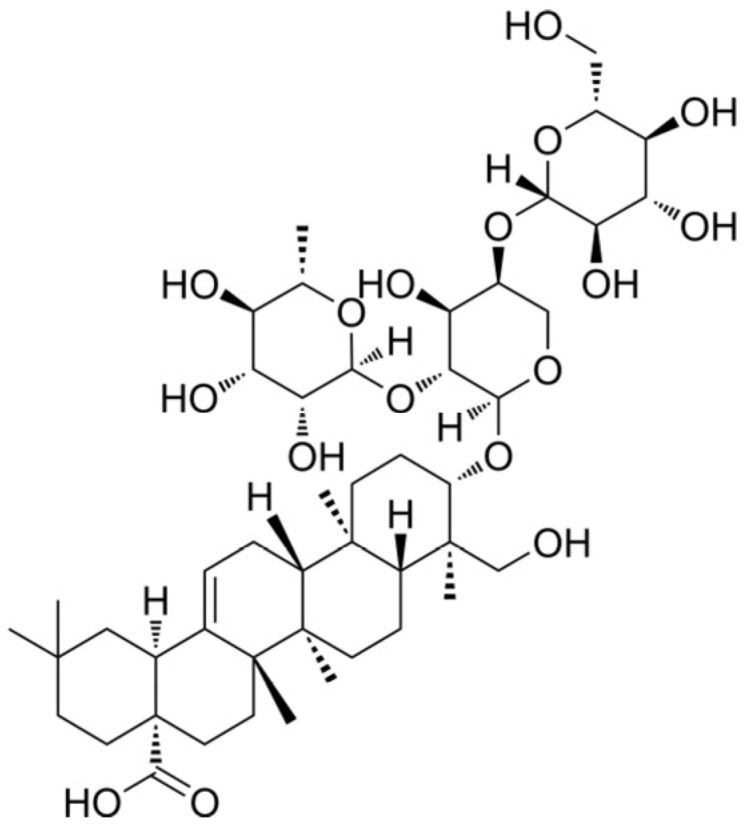
Chemical structure of Pulsatilla saponin D.

**Figure 2 cells-14-01706-f002:**
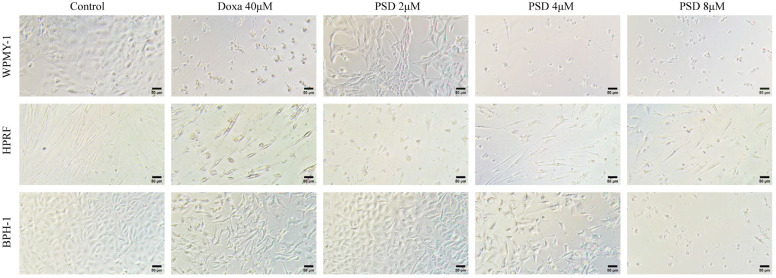
Morphological changes in WPMY-1, HPRF, and BPH-1 cells after treatment with PSD and doxazosin (Doxa) for 48 h (100×, scale bar = 50 μm).

**Figure 3 cells-14-01706-f003:**
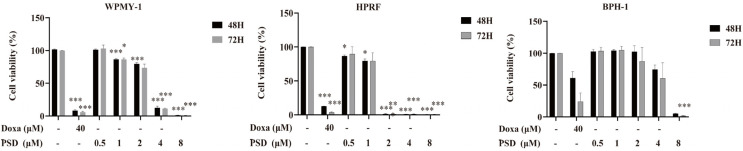
Effects of PSD and doxazosin treatment for 48 and 72 h on the viability of WPMY-1, HPRF, and BPH-1 cells. Data are presented as the mean ± standard deviation (SD) from three independent experiments (*n* = 3). * *p* < 0.05, ** *p* < 0.01, *** *p* < 0.001 versus the respective control group.

**Figure 4 cells-14-01706-f004:**
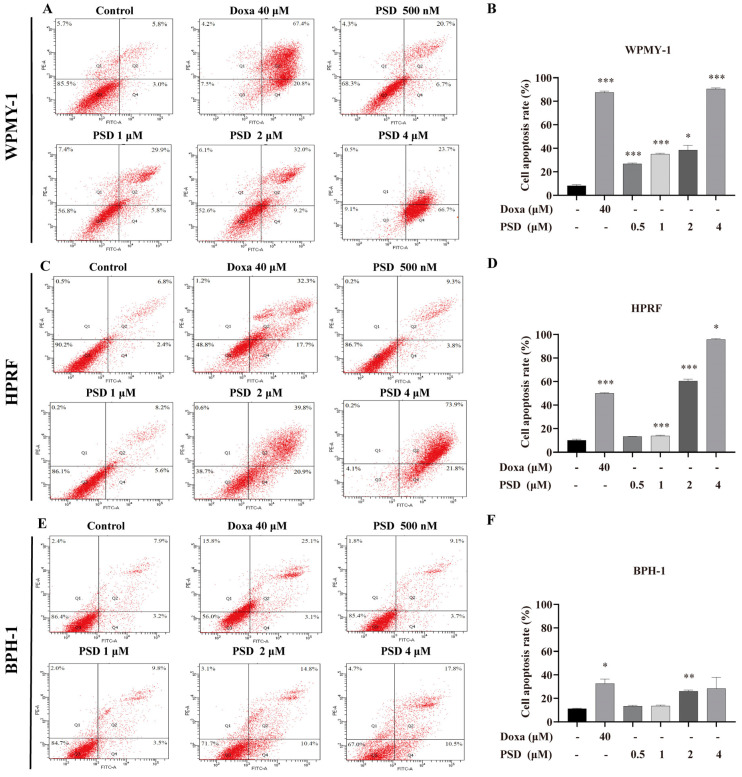
Analysis of apoptosis in WPMY-1, HPRF, and BPH-1 cells after 48 h treatment with PSD. (**A**) Apoptosis scatter plot and (**B**) histogram of apoptosis rate in WPMY-1 cells. (**C**) Apoptosis scatter plot and (**D**) histogram of apoptosis rate in HPRF cells. (**E**) Apoptosis scatter plot and (**F**) histogram of apoptosis rate in BPH-1 cells. Data are presented as the mean ± standard deviation (SD) from three independent experiments (*n* = 3). * *p* < 0.05, ** *p* < 0.01, *** *p* < 0.001 versus the respective control group.

**Figure 5 cells-14-01706-f005:**
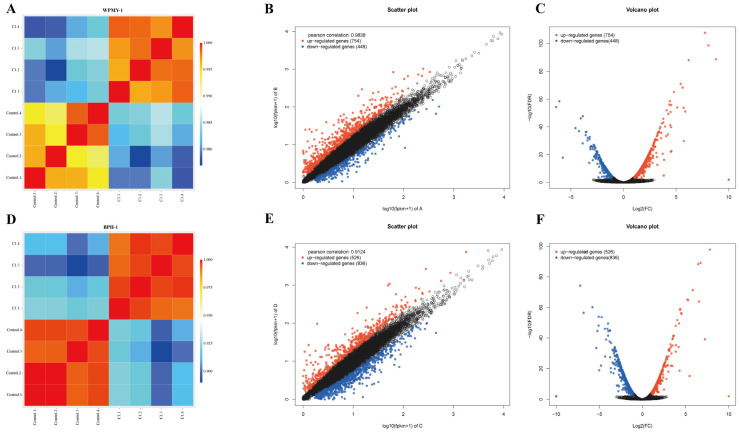
Transcriptomic analysis of WPMY-1 and BPH-1 cells following PSD treatment. (**A**) Sample correlation heatmap of the WPMY-1 cell line. (**B**) Scatter plot of DEGs in WPMY-1 cells. (**C**) Volcano plot displaying DEGs in WPMY-1 cells. (**D**) Sample correlation heatmap of the BPH-1 cell line. (**E**) Scatter plot of differentially expressed genes in BPH-1 cells. (**F**) Volcano plot displaying DEGs in BPH-1 cells.

**Figure 6 cells-14-01706-f006:**
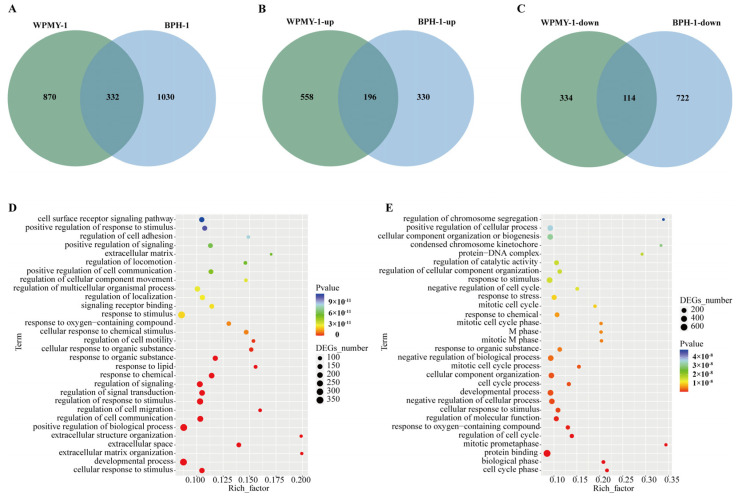
Common differentially expressed genes following PSD treatment. (**A**) Venn diagram of common DEGs shared between WPMY-1 and BPH-1 cells. (**B**) Common up-regulated DEGs. (**C**) Common down-regulated DEGs. (**D**) Gene Ontology (GO) enrichment analysis of DEGs in WPMY-1 cells. (**E**) GO enrichment analysis of DEGs in BPH-1 cells.

**Figure 7 cells-14-01706-f007:**
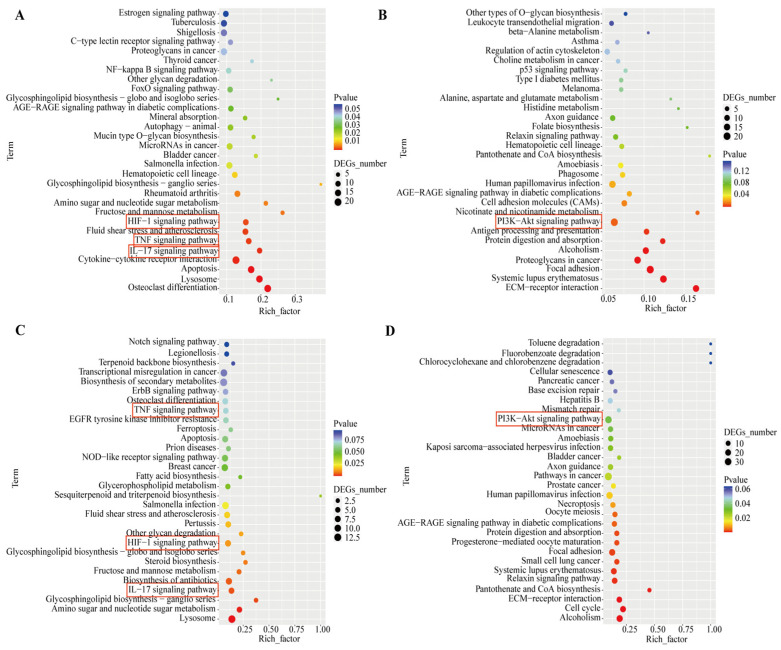
PSD modulates gene expression associated with benign prostatic hyperplasia (BPH). (**A**) KEGG pathway enrichment analysis of significantly up-regulated genes in WPMY-1 cells. (**B**) KEGG pathway enrichment analysis of significantly down-regulated genes in WPMY-1 cells. (**C**) KEGG pathway enrichment analysis of significantly up-regulated genes in BPH-1 cells. (**D**) KEGG pathway enrichment analysis of significantly down-regulated genes in BPH-1 cells.

**Figure 8 cells-14-01706-f008:**
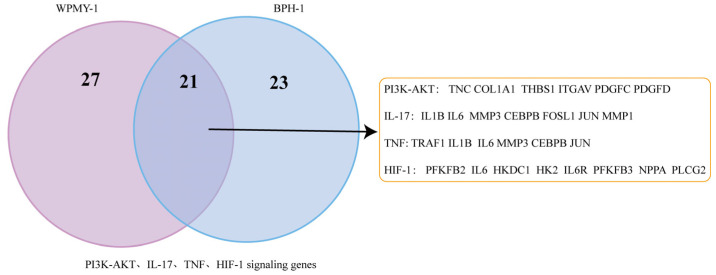
Shared differentially expressed genes in the PI3K/AKT, IL-17, TNF, and HIF-1 signaling pathways between WPMY-1 and BPH-1 cells.

**Figure 9 cells-14-01706-f009:**
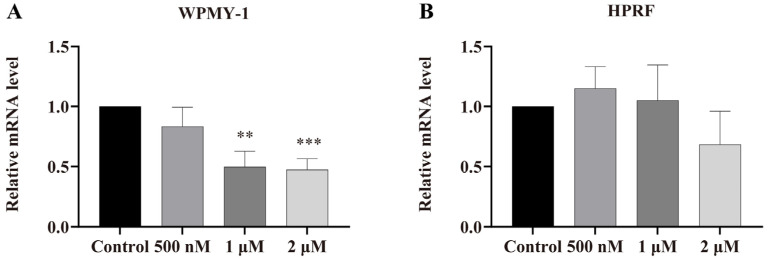
AR Gene Expression Detected by RT-qPCR. (**A**) AR gene expression levels in WPMY-1 cells. (**B**) AR gene expression levels in BPH-1 cells. Data are presented as the mean ± standard deviation (SD) from three independent experiments (*n* = 3). ** *p* < 0.01, *** *p* < 0.001 versus the respective control group.

**Table 1 cells-14-01706-t001:** Primer sequences for RT-qPCR.

Genes	Forward Primer (5′-3′)	Reverse Primer (5′-3′)
*AR*	GTGGACGACCAGATGGCTGTCATTC	GGCGAAGTAGAGCATCCTGGAGTTG

**Table 2 cells-14-01706-t002:** IC_50_ values of PSD in human prostate cell lines WPMY-1, HPRF, and BPH-1.

Cell Lines	IC_50_ (48 h)	IC_50_ (72 h)
WPMY-1	2.649	2.511
HPRF	1.201	1.192
BPH-1	4.816	4.315

## Data Availability

The original contributions presented in this study are included in the article. Further inquiries can be directed to the corresponding author.

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
