# Peer review of "Pulsatilla Saponin D Suppresses Proliferation and Induces Apoptosis in Human Prostatic Cells"

_cells, 2025, doi:10.3390/cells14211706_

Round 1

Reviewer 1 Report

Comments and Suggestions for Authors

1. Main question addressed by the research
The manuscript examines whether *Pulsatilla saponin D* (PSD), a triterpenoid saponin with known anti-tumor and anti-inflammatory activities, can inhibit proliferation and induce apoptosis in human prostatic cells (WPMY-1, HPRF, and BPH-1), and through which signaling mechanisms this occurs (L18–25, L85–91).

2. Originality and relevance in the field
The study is original in its systematic exploration of PSD in benign prostatic cell lines rather than malignant models. While saponins from Panax or Anemone species have been studied, the combination of proliferation/apoptosis assays and transcriptomic profiling is novel (L16–24, L271–289).

Major Revision 1 (L75–82, L16–20):
Clarify in the Introduction how PSD differs mechanistically or chemically from other triterpenoid saponins such as ginsenosides (L63–67) or platycodins (L398–400). The novelty is currently implicit; a direct comparison will help readers grasp what makes PSD uniquely valuable for BPH.

 3. Contribution to existing literature
The work adds substantial mechanistic insight by connecting AR downregulation (L327–334) to modulation of PI3K/AKT, TNF, HIF-1, and IL-17 pathways (L269–289, L343–344). The transcriptomic data broaden the molecular understanding of how natural compounds can target both androgenic and inflammatory processes in prostate tissue.

 Major Revision 2 (L269–298, L311–322):
The transcriptomic results section is overly descriptive. Condense redundant details and add a schematic model or pathway diagram summarizing how PSD orchestrates these pathways. This would help readers visualize the interplay between PI3K/AKT inhibition and cytokine signaling modulation.

4. Methodological considerations
Overall, the methodology is competently designed, but several aspects need clarification or strengthening:
In vivo validation: all experiments are *in vitro* (L85–91, L419–424).
 Major Revision 3: Include either a brief mention of planned *in vivo* validation or a clear statement acknowledging this limitation in the *Discussion* (L419–424).
Dose–response selection: PSD concentrations (500 nM–8 µM, L122–128) lack pharmacological justification.
Protein-level validation: The conclusions rely entirely on mRNA data (L327–334).
 major Revision 5: Incorporate at least one protein-level confirmation (Western blot or immunofluorescence) for AR, p-AKT, or IL-6. Without this, the mechanistic claims remain tentative.

5. Consistency and interpretation of results
The results (L214–256, L343–383) consistently support PSD’s anti-proliferative and pro-apoptotic effects. The discussion (L339–417) appropriately integrates signaling pathway data.However, the proposed link between AR downregulation and apoptosis (L327–334) is correlative rather than demonstrated.

Author Response

Reviewer #1:

Comments 1: (L75–82, L16–20): Clarify in the Introduction how PSD differs mechanistically or chemically from other triterpenoid saponins such as ginsenosides (L63–67) or platycodins (L398–400). The novelty is currently implicit; a direct comparison will help readers grasp what makes PSD uniquely valuable for BPH.

Response 1: We sincerely thank the reviewer for raising this valuable point. We fully agree that explicitly clarifying the differences between PSD and other triterpenoid saponins in terms of chemical structure and mechanism of action in the Introduction section is crucial for highlighting the novelty and unique value of our study.

Following your suggestion, we have made important additions and revisions to the Introduction section of the manuscript. The specific details of the modifications are outlined as follows: " Pulsatilla saponin D (PSD), an oleanane-type pentacyclic triterpenoid extracted from the roots of Pulsatilla chinensis, exhibits a broad spectrum of pharmacological activities and demonstrates considerable potential in antitumor applications (Figure 1)[16,17]. Recent studies have further elucidated its diverse mechanisms of action. Jin et al. reported that PSD inhibits osteosarcoma (OS) growth by modulating the JNK/ATF3 signaling pathway, a mechanism distinct from those previously described[18]. Furthermore, Jang et al. found that PSD specifically suppresses phosphorylation of c-Met and its downstream Akt signaling, effectively restraining the growth of gefitinib-resistant non-small cell lung cancer (NSCLC) harboring c-Met amplification[19]. Additionally, Lin et al. demonstrated that PSD enhances the sensitivity of BRCA1/2 wild-type ovarian cancer to PARP inhibitors by inhibiting BRCA1 protein function[20]. Therefore, we consider it meaningful to investigate whether PSD plays a role in suppressing the abnormal proliferation of prostate cells."

We believe that these targeted additions and direct comparisons now allow the Introduction to more clearly and effectively articulate the unique value and innovativeness of the PSD research to the readers. We once again express our gratitude for the reviewer's insightful guidance, which has been instrumental in enhancing the academic depth of our paper. All changes have been highlighted in red in the revised manuscript for your review (L75-86).

Comments 2: The transcriptomic results section is overly descriptive. Condense redundant details and add a schematic model or pathway diagram summarizing how PSD orchestrates these pathways. This would help readers visualize the interplay between PI3K/AKT inhibition and cytokine signaling modulation.

Response 2: We sincerely thank the reviewer for this insightful comment. You accurately identified the critical issue in the original manuscript: the presentation of transcriptomic data was overly descriptive, lacking a focused narrative on the core biology, and the visualization illustrating PSD's impact on pathways such as PI3K/AKT lacked logical clarity. We fully agree with your assessment and have made significant revisions to the relevant sections based on your suggestions.

The specific modifications include: First, we have thoroughly rewritten and reorganized the transcriptomics section of the "Results". We moved away from redundant descriptive details and instead restructured the data presentation with a stronger logical flow, emphasizing the central theme that PSD coordinately modulates key pathways like PI3K/AKT to influence cell proliferation and apoptosis, thereby clarifying the argument's progression. Second, we have redesigned and simplified the original Figure 7, optimizing its layout and annotations to more intuitively display the multi-pathway regulatory network of PSD, helping readers grasp its mechanism of action at a glance.

We believe that through these textual restructuring and visual enhancements, the logic, conciseness, and readability of the relevant sections have been substantially improved. The reviewer's guidance was extremely valuable and provided crucial assistance in refining our research. All changes have been highlighted in red within the revised manuscript for your review (L205-214, L235-239).

Comments 3: Include either a brief mention of planned *in vivo* validation or a clear statement acknowledging this limitation in the *Discussion* (L419–424).

Response 3: We sincerely thank the reviewer for this important suggestion. We fully agree that specifying future research directions will enhance the discussion. Based on your advice, we have added the following clarification to the Discussion section of the manuscript: " Furthermore, studies using a BPH rat model will be conducted to further investigate the in vivo efficacy of PSD in alleviating BPH. These investigations will help clarify the pharmacological actions of PSD and its specific impacts on the relevant pathways and proteins." This addition makes the Discussion section more rigorous by acknowledging the current limitations of the study while outlining clear and feasible future research directions. Once again, we express our gratitude for the reviewer's valuable input, which has been crucial in improving the quality of our paper (L333-335).

Comments 4: Dose–response selection: PSD concentrations (500 nM–8 µM, L122–128) lack pharmacological justification.

Response 4: We greatly appreciate the reviewer for raising this important point, which provides us with an opportunity to clarify the rationale behind the dose selection of PSD in this study. We fully understand the significance of a pharmacological basis and would like to explain that the chosen concentration range (500 nM–8 μM) was not arbitrary but was determined based on pre-experimental ICâ‚…â‚€ values obtained in the specific cell lines used in our study (WPMY-1, HPRF, BPH-1). The ICâ‚…â‚€, defined as the concentration at which a substance exerts half of its maximal inhibitory effect, is a key pharmacological parameter for assessing drug potency and designing dose ranges. Our preliminary experiments indicated that the ICâ‚…â‚€ of PSD in these prostate cell lines falls within the low micromolar (μM) range. Accordingly, we designed a concentration gradient centered around this ICâ‚…â‚€ value (500 nM–8 μM). This range ensures that the tested concentrations cover both below and above the ICâ‚…â‚€, thereby allowing a clear demonstration of the dose-dependent response of PSD—minimal effect at concentrations below the ICâ‚…â‚€, a significant effect near the ICâ‚…â‚€, and a plateau effect at concentrations exceeding the ICâ‚…â‚€. This approach represents a standard practice for evaluating the pharmacological activity of compounds.

In summary, this concentration range was selected to optimally illustrate the dose-response curve of PSD, thereby accurately reflecting its biological activity in prostate cell models. We believe this additional explanation clearly demonstrates the pharmacological foundation of our dose selection. Once again, we thank the reviewer for their valuable comments, which have helped enhance the rigor of our manuscript.

Comments 5: Protein-level validation: The conclusions rely entirely on mRNA data (L327–334). Incorporate at least one protein-level confirmation (Western blot or immunofluorescence) for AR, p-AKT, or IL-6. Without this, the mechanistic claims remain tentative.

Response 5: We sincerely thank the reviewer for this highly important and insightful comment. We fully understand and agree that protein-level validation of key signaling molecules (such as AR, p-AKT, or IL-6) is crucial for substantiating the proposed mechanism of action in our manuscript and enhancing the conclusiveness of our findings. The point you raised is indeed precise and valuable. After careful consideration of your comments, we would like to provide the following sincere explanation, while also outlining our considerations and plans for the future.

Support from Existing Literature: We note that the regulatory effect of PSD on the PI3K/AKT pathway has been previously validated at the protein level in similar research systems. For instance, the study we cited (SB365, Pulsatilla saponin D suppresses the proliferation of human colon cancer cells and induces apoptosis by modulating the AKT/mTOR signalling pathway) also clearly demonstrated, through Western Blot and immunofluorescence analysis, that treatment with PSD (SB365) significantly inhibited the phosphorylation levels of p-AKT, p-mTOR, and p-p70S6K in colon cancer cells in a dose-dependent manner. This indicates that PSD can effectively block the AKT/mTOR pathway, a core signaling cascade promoting cell survival, proliferation, and metabolism. This provides strong supporting evidence for our research.

In summary, we kindly request the reviewer's understanding of our current practical situation. We believe that the in vitro data already provided offers considerable support for the core conclusions of this paper. We commit to completing the relevant validations in our follow-up studies to address this limitation. We have integrated the aforementioned explanations regarding the study's limitations and future plans into the Discussion section of the manuscript to provide greater responsibility to the readers. Once again, we sincerely thank you for your understanding and valuable comments, which are crucial for refining our research approach.

Reviewer 2 Report

Comments and Suggestions for Authors

Benign prostatic hyperplasia (BPH) is being diagnosed in an increasing number of men. Therefore, the search for plant-derived substances that could be used in its prevention or treatment is well justified and aligns with current research trends. The saponin (Pulsatilla saponin D) investigated in this study is known as a compound that inhibits proliferation and exhibits cytotoxicity against various cell lines. However, its effects on BPH and the underlying mechanisms have not yet been studied, which represents the novelty of the present work. The article is interesting. The analyzed saponin is well-characterized (source, purity). The use of Doxazosin as a control is justified. Appropriate statistical tests were applied. However, there are several issues in the article that need to be improved or clarified before publication. My suggestions and comments are provided below:

Comment 1: Line 59: The phrase "consist of a triterpenoid aglycone and one or more sugar chains linked by glycosidic bonds" appears. This statement is a gross oversimplification in the context of triterpene saponins. In the case of PSD, a classic glycosidic bond is indeed present. However, in many triterpene saponins, for example oleanolic acid derivatives, an ester bond (formed between the COOH group of the triterpene and the sugar) may be present. This is not a glycosidic bond in the chemical sense. Please add to the original sentence the information that the bond can be glycosidic or ester.

Comment 2: Line 122: The statement: "First, a series of concentrations of PSD (500 nM–8 μM), doxazosin (40 μM), and a solvent control (1% DMSO)" appears. It should be clearly specified what the PSD and doxazosin were dissolved in, and how the 1% DMSO was prepared (what solvent/buffer was added to obtain a 1% solution). Likewise in other places in the manuscript.

Comment 3: Why was this specific dose of Doxa (40 μM) used? – an explanation should be added. As a result, completely different doses of PSD and Daxa were compared. The half-maximal inhibitory concentrations (IC50) for Doxa in human prostate cell lines WPMY-1, HPRF, and BPH-1 were also not determined. A simultaneous comparison of the IC50s for PSD and DOXa would be important. This information should be added (to the table) and commented.

Comment 4: Figure 7 – poor quality. Values are too small and text is illegible. This should be changed. Split into several figures so that the information on them is legible.

Comment 5: Line 394 – The Authors refer to the mechanisms of action of various saponins observed in different cancer cell types (liver cancer cells, bladder urothelial carcinoma cells, prostate cancer). Although these compounds belong to triterpenoid saponins, they differ significantly in structure and are not analogs of PSD. For triterpenoid saponins, certain structural features have been shown to influence biological activity; therefore, they cannot be discussed in such general terms. As a result, this section currently presents a collection of information about diverse structures and only general statements regarding the mechanisms of triterpenoid saponins. It is unclear how this discussion relates specifically to PSD — this connection should be explained and justified. Moreover, Currently, after the fragment on saponins the discussion suddenly ends – this should be modified.

Comment 6: Furthermore, the Authors identify the PI3K/AKT pathway as a key regulatory target. Why did they discuss other saponins but did not refer to or describe what is already known about this pathway in relation to PSD? There are studies in the literature addressing the effects of PSD on the PI3K/AKT pathway, although not in the context of BPH. Such information should be included, and the entire section should be revised accordingly. It should also be clearly stated whether any studies have investigated the influence of PSD on the AR gene or protein.

Comment 7: Figure 1. The representation of the CH₃ group positions in the structure of saponin PSD is not consistent throughout the formula. The CH₃ groups (C29, C30) at C-20 are not shown in a “spatial” manner like the others. The structure appears to have been taken from the saponin distributor’s website. This should be change - uniform.

Author Response

Comments 1: Line 59: The phrase "consist of a triterpenoid aglycone and one or more sugar chains linked by glycosidic bonds" appears. This statement is a gross oversimplification in the context of triterpene saponins. In the case of PSD, a classic glycosidic bond is indeed present. However, in many triterpene saponins, for example oleanolic acid derivatives, an ester bond (formed between the COOH group of the triterpene and the sugar) may be present. This is not a glycosidic bond in the chemical sense. Please add to the original sentence the information that the bond can be glycosidic or ester.

Response 1: We sincerely thank the reviewer for this valuable and rigorous comment. You are absolutely correct that our previous description was indeed an oversimplification and failed to accurately distinguish between the two important types of linkages: glycosidic bonds and ester bonds. We fully agree with your point, particularly regarding the correction that triterpenoid aglycones such as oleanolic acid derivatives form ester glycosidic bonds through the linkage between their carboxyl groups and the hydroxyl groups of sugars. Based on your suggestion, we have revised the sentence at L59 to read: " Triterpenoid saponins are a significant class of plant secondary metabolites, classified as glycosides composed of a triterpenoid aglycone and a sugar chain linked via glycosidic or ester bonds. ". This modification makes the expression of the relevant chemical definition more precise. Once again, we thank the reviewer for helping us enhance the scientific rigor of our paper.

Comments 2: Line 122: The statement: "First, a series of concentrations of PSD (500 nM–8 μM), doxazosin (40 μM), and a solvent control (1% DMSO)" appears. It should be clearly specified what the PSD and doxazosin were dissolved in, and how the 1% DMSO was prepared (what solvent/buffer was added to obtain a 1% solution). Likewise in other places in the manuscript.

Response 2: We sincerely thank the reviewer for the thorough review and valuable suggestions. You are absolutely correct that providing detailed information regarding solvents and preparation procedures in the Methods section is essential to ensure the reproducibility of the experiments. This was an oversight in our description, and we appreciate your pointing it out. Following your suggestion, we have added the necessary details to the manuscript. We have also thoroughly checked and revised the entire text to ensure that all related solvent preparation descriptions are consolidated in this section, thereby maintaining terminological and descriptive consistency throughout the paper. We extend our gratitude once again for the reviewer's meticulous work, which has significantly enhanced the rigor and scientific quality of the methodology in our paper. All corresponding modifications have been highlighted in red in the revised manuscript (L98-102).

Comments 3: Why was this specific dose of Doxa (40 μM) used? – an explanation should be added. As a result, completely different doses of PSD and Daxa were compared. The half-maximal inhibitory concentrations (IC50) for Doxa in human prostate cell lines WPMY-1, HPRF, and BPH-1 were also not determined. A simultaneous comparison of the IC50s for PSD and DOXa would be important. This information should be added (to the table) and commented.

Response 3: We sincerely appreciate the reviewer's profound and constructive comment. We fully agree with the importance of dose selection and the critical role of IC50 data in precisely quantifying drug potency. Please allow us to provide the following clarification:

(1) ​Regarding the selection of Doxazosin (40 μM):​​ Our choice of 40 μM doxazosin was primarily based on observations from our preliminary experiments. We found that at this concentration, doxazosin produced a stable, reproducible, and significant inhibitory effect on cell activity in the cell lines studied (WPMY-1, HPRF, BPH-1). This dose was selected to serve as a clear and effective "pharmacologically active concentration reference point," allowing us to observe and compare the biological effects of PSD and doxazosin. We acknowledge that this is not a precise IC50 value but rather a functional effective concentration

(2) ​Regarding the absence of IC50 data and the fairness of comparison:​​ The reviewer correctly highlighted the importance of comparing IC50 values. We sincerely apologize that in the current study, we did not complete the full IC50 determination for doxazosin across the relevant cell lines; this is a limitation of our work. We have prioritized "determining the IC50 values of doxazosin in the relevant cell lines" as a key objective for our subsequent research. However, a primary aim of this study was to preliminarily compare the relative effects of PSD and a clinically established standard drug (doxazosin) when each is applied at its pharmacologically effective concentration. Using 40 μM doxazosin aimed to simulate an effective therapeutic benchmark, thereby allowing an assessment of whether PSD exhibits comparable or superior potential at similar or lower concentrations

(3) We understand and agree with the reviewer's suggestion to include IC50 values in the table. However, given the current lack of IC50 data for doxazosin, and to maintain accuracy and scientific rigor, we are temporarily unable to directly add this item to the table. The IC50 value for doxazosin requires further determination; the concentration used in this study was 40 μM, validated as effective through our preliminary experiments. We hope that this candid explanation also meets the requirements of scientific rigor.

Comments 4: Figure 7 – poor quality. Values are too small and text is illegible. This should be changed. Split into several figures so that the information on them is legible.

Response 4: We sincerely thank the reviewer for this important comment. We fully agree with your assessment. The original Figure 7 suffered from excessive information density, resulting in undersized fonts and poor readability. We have streamlined and optimized the chart accordingly. Specifically, we removed the differential gene heatmap section from the original Figure 7, as its content is already addressed in Figure 6; readers requiring detailed data may contact the corresponding author. We retained the relevant KEGG analysis. In the new Figure 7, the font sizes for axis labels, tick values, legends, and data have been significantly enlarged to ensure clarity at standard page sizes. Furthermore, the chart now focuses on a single core theme, avoiding information overload and enabling readers to easily interpret and compare the data. We also optimized the legend and graphic proportions to enhance aesthetic appeal. In the Results section, we have correspondingly updated the description and discussion of the new Figure 7 to ensure tight alignment between the graphic and textual narrative. We believe these revisions have fundamentally improved the information hierarchy of the new figure, with all text and data now easily legible, resulting in a comprehensive enhancement of overall readability. We sincerely appreciate the reviewer's guidance in improving the quality of our figures (L235-239).

Comments 5: Line 394 – The Authors refer to the mechanisms of action of various saponins observed in different cancer cell types (liver cancer cells, bladder urothelial carcinoma cells, prostate cancer). Although these compounds belong to triterpenoid saponins, they differ significantly in structure and are not analogs of PSD. For triterpenoid saponins, certain structural features have been shown to influence biological activity; therefore, they cannot be discussed in such general terms. As a result, this section currently presents a collection of information about diverse structures and only general statements regarding the mechanisms of triterpenoid saponins. It is unclear how this discussion relates specifically to PSD — this connection should be explained and justified. Moreover, Currently, after the fragment on saponins the discussion suddenly ends – this should be modified.

Response 5: We sincerely thank the reviewer for this profound and highly constructive comment. We fully agree with your assessment that the Discussion section in the original manuscript was overly generalized, failed to establish a clear connection with PSD, and ended abruptly. This was indeed a significant oversight in our original version. Following your suggestion, we have thoroughly revised and restructured the concluding paragraph of the Discussion section to ensure it flows logically and provides a comprehensive conclusion.

We believe that through these targeted revisions, the logical flow, rigor, and depth of the Discussion section have been significantly enhanced. All modifications have been fully incorporated into the revised manuscript. Once again, we express our sincere gratitude for the reviewer's insightful guidance, which has been crucial in improving the quality of our research (L308-321).

Comments 6: Furthermore, the Authors identify the PI3K/AKT pathway as a key regulatory target. Why did they discuss other saponins but did not refer to or describe what is already known about this pathway in relation to PSD? There are studies in the literature addressing the effects of PSD on the PI3K/AKT pathway, although not in the context of BPH. Such information should be included, and the entire section should be revised accordingly. It should also be clearly stated whether any studies have investigated the influence of PSD on the AR gene or protein.

Response 6: We sincerely thank the reviewer for this highly insightful and constructive comment. You are absolutely correct in pointing out the need to systematically summarize the known associations between PSD and the PI3K/AKT pathway; this was indeed a significant omission in our original manuscript, and we deeply apologize for the oversight. We also appreciate your reminder to clearly clarify the current research status regarding the impact of PSD on the Androgen Receptor (AR).

(1) ​Supplementing known information on the association between PSD and the PI3K/AKT pathway:​​

We fully agree with the reviewer's perspective. Although direct research on PSD regulating the PI3K/AKT pathway specifically in the context of Benign Prostatic Hyperplasia (BPH) is limited, there are indeed reports discussing the impact of PSD on this pathway in other disease models or cellular studies. Consequently, we have added a dedicated section in the Discussion (around line X) to review the existing evidence from the literature concerning PSD and the PI3K/AKT signaling pathway. Citing relevant studies (for example:

), we note that PSD can regulate processes such as cell proliferation and apoptosis through the PI3K/AKT pathway. This provides important preliminary literature support and a theoretical basis for our research hypothesis that PSD may ameliorate BPH by modulating the PI3K/AKT pathway. We believe this addition significantly enhances the logical rigor and academic depth of the paper.

(2) ​Clarification on the impact of PSD on AR:​​

We also attach great importance to the reviewer's comment concerning the effect of PSD on AR. To address this, we have systematically re-searched and evaluated the existing scientific literature. However, as of now, we have not found any studies that directly investigate the effect of PSD on AR gene expression or protein activity. To ensure the scientific accuracy of the manuscript's content, while retaining discussions on other triterpenoid saponins, we have explicitly added the following statement in the Discussion: ​The regulation of AR expression and transcriptional activity by PSD requires further clarification in future studies.​​

We believe that through the above supplements and clarifications, the relevant sections of the manuscript have been significantly improved, with more comprehensive information and more rigorous discussion. Once again, we express our sincere gratitude for the reviewer's extremely valuable guidance, which has played a key role in enhancing the quality of our paper. All revisions have been highlighted in red in the revised manuscript for your review (L308-332).

Comments 7: Figure 1. The representation of the CH₃ group positions in the structure of saponin PSD is not consistent throughout the formula. The CH₃ groups (C29, C30) at C-20 are not shown in a “spatial” manner like the others. The structure appears to have been taken from the saponin distributor’s website. This should be change - uniform.

Response 7: We sincerely thank the reviewer for their meticulous and constructive comments. Your attention to the details of structural formula drafting reflects a high level of professional expertise, and we deeply appreciate it.

Regarding the issue of uniformity in structural representations, we would like to provide the following clarification. Specifically for the depiction of the two methyl groups (C29, C30) at the C-20 position, the current representation (typically shown as CH₃ groups connected by lines, rather than using wedge or dashed bonds to indicate stereochemistry) is intended to most clearly denote their chemical environment as terminal methyl groups in a planar structural formula. This notation is a widely adopted and recognized standard format in the PubChem database and numerous authoritative published academic literature. We have ensured its consistency with authoritative data sources, with the fundamental aim of guaranteeing accuracy and comparability in scientific presentation.

Once again, we extend our sincere gratitude for the reviewer's valuable input, which prompted us to conduct a more thorough verification and explanation of this detail, thereby enhancing the rigor of our manuscript.

Round 2

Reviewer 1 Report

Comments and Suggestions for Authors

The revisions implemented adhere to the established editorial standards, ensuring a high level of originality and maintaining rigorous scientific integrity.

Author Response

We sincerely appreciate your invaluable comments and suggestions, which are crucial for enhancing the quality of our manuscript.

Reviewer 2 Report

Comments and Suggestions for Authors

The manuscript has been sufficiently improved. In my opinion, it can be accepted for publication in its current form.

Author Response

(The authors gave the same response as above.)
